

# A tensor network study of the complete ground state phase diagram of the spin-1 bilinear-biquadratic Heisenberg model on the square lattice

**Ido Niesen[1][⋆] and Philippe Corboz[1]**

**1** Institute for Theoretical Physics and Delta Institute for Theoretical Physics, University of Amsterdam, Science Park 904, 1098 XH Amsterdam, The Netherlands

⋆ i.a.niesen@uva.nl

## Abstract

Using infinite projected entangled pair states, we study the ground state phase diagram of the spin-1 bilinear-biquadratic Heisenberg model on the square lattice directly in the thermodynamic limit. We find an unexpected partially nematic partially magnetic phase in between the antiferroquadrupolar and ferromagnetic regions. Furthermore, we describe all observed phases and discuss the nature of the phase transitions involved.

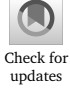

# 1 Introduction

Strongly correlated materials—even those described by relatively simple Hamiltonians—can display very rich behavior. In frustrated magnets, for example, classical ground state[1] degeneracies due to the inability of the system to minimize each term in the Hamiltonian individually give way for quantum fluctuations to play the predominant role in determining the actual ground state of the system. As such, frustrated magnets are a playground for all kinds of exotic states of matter, such as spin liquids [1, 2] and spin-nematic states [3, 4], the latter of which break spin-rotational symmetry while preserving time-reversal symmetry.

In this paper, we study the spin-1 bilinear-biquadratic Heisenberg (BBH) model on the square lattice: a model that consists of spin-1 particles, one per lattice site, that interact only with their nearest neighbors. The nearest-neighbor two-particle interaction is a combination of two types of competing interactions: the ordinary bilinear Heisenberg coupling, and the biquadratic coupling, which is the Heisenberg coupling squared. The corresponding coupling constants appearing in the Hamiltonian are commonly parametrized by an angle $\theta$:

$$H = \sum_{\langle i,j \rangle} \cos(\theta)\, \boldsymbol{S}_i \cdot \boldsymbol{S}_j + \sin(\theta)\big(\boldsymbol{S}_i \cdot \boldsymbol{S}_j\big)^2, \tag{1}$$

where $\boldsymbol{S}_i = (S_i^x, S_i^y, S_i^z)$ is the vector of spin-matrices for the spin-1 particle on site $i$,

$$S_i^x = \frac{1}{\sqrt{2}}\begin{pmatrix} 0 & 1 & 0 \\ 1 & 0 & 1 \\ 0 & 1 & 0 \end{pmatrix}, \quad S_i^y = \frac{i}{\sqrt{2}}\begin{pmatrix} 0 & -1 & 0 \\ 1 & 0 & -1 \\ 0 & 1 & 0 \end{pmatrix}, \quad \text{and} \quad S_i^z = \begin{pmatrix} 1 & 0 & 0 \\ 0 & 0 & 0 \\ 0 & 0 & -1 \end{pmatrix},$$

and the sum goes over all nearest-neighbor pairs.

The BBH model on the square lattice has been subject to a lot of interest in recent years, for a number of reasons. For one, at $\theta = \pi/4$, the Hamiltonian is equivalent to the SU(3) Heisenberg model [5, 6], which can be realized using cold-atom experiments [7–10]. Second, it was proposed that the nematic phases of the BBH model on the triangular lattice could be related to the unusual behavior of $NiGa_2S_4$ [11–13] and $Ba_3NiSb_2O_9$ [14–16].

From a theoretical perspective, the BBH model is the most general lattice-translation, lattice-rotation and spin-rotation-symmetric Hamiltonian with nearest-neighbor interactions.[2]

---

[1]By a *classical ground state* we mean a lowest energy state within the manifold of site-factorized product states.
[2]For spin-1 particles, all higher powers of $\boldsymbol{S}_i \cdot \boldsymbol{S}_j$ can be rewritten in terms of the bilinear and biquadratic terms.

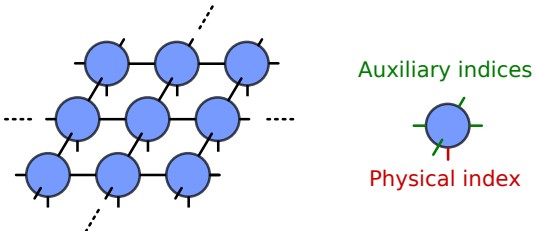

Figure 1: Left: part of an iPEPS for a square lattice. Right: physical and auxiliary indices; each leg corresponds to an index. Connected lines are summed over.

Moreover, the more interesting part of the phase diagram—the top half $0 < \theta < \pi$—is beyond the reach of quantum Monte Carlo [17] because of the sign problem, and has up to now only been studied either for small system sizes using exact diagonalization [18], or on larger systems using semi-classical [4,18] and perturbative [19] approaches.

In this article we continue upon our work on the emergence of the Haldane phase in the spin-1 BBH model on the square lattice [20] and provide a complete study of the ground state phase diagram using state-of-the-art tensor network algorithms. Contrary to previous studies, we find the occurrence of a half nematic half magnetic '$m = 1/2$' phase that was previously predicted to appear only in the presence of an external magnetic field [18]. Additionally, we describe all phases and determine the nature of the corresponding phase transitions. The ground state phase diagram of the spin-1 BBH model according to our computations, which in itself is the main results of this paper, can be found in Fig. 8.

This paper is organized as follows. In Section 2 we provide a brief description of the numerical method used. Thereafter, Section 3 sets the stage by discussing some of the relevant properties of spin-1 particles—in particular in the context of spin-nematic order—as well as the symmetries of the Hamiltonian, followed by an overview of previous studies of the BBH model in Section 4. We then proceed to discuss our own findings and compare those with previous results in Section 5, and conclude with Section 6.

## 2 Method

We simulate the ground state of the spin-1 BBH model for different values of the coupling parameter $\theta$ using infinite projected entangled pair states [21,22] (iPEPS), and then optimize the iPEPS tensors via imaginary time evolution to find an approximate ground state of the system. iPEPS is the infinite-system version of PEPS [23] (also called a *tensor product state* [24,25]), the latter of which is a two-dimensional generalization of matrix product states (MPS) [26–28]: the ansatz underlying the well-known density-matrix-renormalization-group (DMRG) algorithm [29–31]. iPEPS has already successfully been applied to a number of strongly correlated two-dimensional systems: see e.g. Refs. [20,32–47].

An iPEPS consists of five-legged tensors, one per lattice site, each with four *auxiliary* legs connecting to the four neighboring tensors in accordance with the square lattice structure, and one *physical* leg corresponding to the local Hilbert space of a single spin-1 particle (see Fig. 1). The auxiliary vector spaces all have the same dimension, called the *bond dimension* ($D$). The bond dimension controls the accuracy of the ansatz: the higher $D$, the more entanglement can be captured by the iPEPS, yielding better approximations to the ground state as $D$ increases. In practice, we run simulations up to at most $D = 10$ ($D = 16$ if the state has U(1)-spin-rotation symmetry) and then extrapolate $D \to \infty$, or, equivalently, truncation error to zero [48], when computing expectation values.

If the ground state is translationally invariant, the iPEPS can be represented by a single tensor repeated all over the lattice. However, if translational invariance is (partially) broken, then we have to use an iPEPS with a larger unit cell to reproduce the ground state; e.g., the antiferromagnetic ground states require a 2x2 unit cell with two different tensors.[3]

One of the main obstacles in dealing with infinite two-dimensional tensor networks is that, contrary to the one-dimensional case, it is impossible to contract the infinite network exactly. Instead, several approximation schemes have been developed over the last years, such as: transfer-matrix-based contraction schemes [21], coarse-graining-based contraction schemes [49,50] and corner-transfer matrix (CTM) methods [51–53] based on a formalism derived by Baxter [54,55]. For this paper, we have used a modified version [40,56] of the CTM algorithm developed by Orús and Vidal [57].

In the CTM contraction scheme, for each tensor in the unit cell additional environment tensors are introduced that represent the contraction of all other tensors surrounding the tensor in question. These environment tensors come with their own environment bond dimension $\chi$, and are obtained as a fixed point of a row and column insertion and truncation procedure [57]. For this paper, we have chosen $\chi(D) > D^2$ to be large enough to yield negligible variations in energy compared to those due to the use of finite $D$.

Given a randomly initialized iPEPS $|\psi\rangle$ with a predefined unit cell, we evolve $|\psi\rangle$ in imaginary time by applying $\exp(-\tau H)$ for large enough $\tau$ to project it onto the ground state. At the cost of a controllable Trotter error, using a second order Trotter decomposition the imaginary time evolution operator $\exp(-\tau H)$ can be decomposed into a sequence of gates that evolve a single bond of $|\psi\rangle$ a small step in imaginary time. Application of each such gate increases the bond dimension of the updated bond. Truncating the bond dimension back to $D$ after updating a single bond can be done using the *simple* or the *full* update algorithm.

The simple update algorithm [50] is inspired by its one-dimensional counterpart, known as infinite time-evolving block decimation (iTEBD) [58]. In one dimension, cutting a single bond corresponds to splitting the Hilbert space into two, and truncating the bond dimension back to $D$ amounts to keeping the $D$ largest Schmidt values corresponding to the cut in question, a procedure that yields the best rank-$D$ approximation to the updated state in the 2-norm. Similarly, the simple update keeps the $D$ largest values of a singular value decomposition involving only the tensors and accompanying weights connected to the to-be-truncated bond, which in this case does not correspond to an actual Schmidt decomposition, because cutting a single bond does not split the Hilbert space into two. Regardless, the simple update, albeit being somewhat ad hoc, is computationally very efficient and yields reasonably accurate results in many cases.

In the full update scheme [22] the truncated tensors are found by variationally minimizing the squared distance to the iPEPS with the updated bond. This requires computing the environment at every step, unless the *fast full update* [59] is used, which speeds up the algorithm by recycling previous environments. The full update algorithm is computationally more costly than its simple counterpart, but it does not suffer from uncontrollable approximations, and it is systematic, in the sense that in the $D \to \infty$ limit the iPEPS should converge to the true ground state of the system in question.

---

[3]Throughout this paper, with 'unit cell' we mean the iPEPS unit cell used in the simulations, not the minimal unit cell that generates the (colored) Bravais lattice—which in the case of AFM would be a 2x1 unit cell, not a 2x2 unit cell.

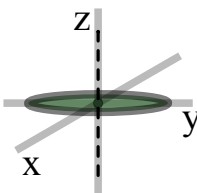

Figure 2: The $|0\rangle$ quadrupolar state. The disc represents the magnetic dipole fluctuations in the x-y plane, and the dotted line represents a director perpendicular to the plane of fluctuations pointing along the z-axis.

## 3 The model

### 3.1 Spin-1 nematic order

Individual spin-1 particles can display more complex behavior than their spin-1/2 counterparts. For example, every spin-1/2 single-particle state—assuming it is a pure state, which we take all single-particle states in this subsection to be—has a fixed magnetization $m = \sqrt{\langle S^x \rangle^2 + \langle S^y \rangle^2 + \langle S^z \rangle^2}$ of exactly 1/2 (setting $\hbar = 1$), and is fully described by its magnetic dipole moment $\boldsymbol{S} = (S^x, S^y, S^z)$. This is a consequence of the fact that any spin-1/2 single-particle state can be obtained by applying a specific rotation to the up state in the z-basis (or any other reference state for that matter). By contrast, spin-1 particles can have a magnetization of anywhere between 0 and 1. Moreover, since spin-1 particles live in a higher-dimensional space, the dipole moment is not sufficient to fully describe a given spin-1 single-particle state, for which in addition also the quadrupole moment is required.

The quadrupole moment is measured by products of spin operators, which we can combine into the matrix $T^{\alpha\beta} = S^\alpha S^\beta$, with $\alpha, \beta \in \{x, y, z\}$. However, $\boldsymbol{T}$ contains more than just the quadrupole moment, as the anti-symmetric part of $\boldsymbol{T}$ is just the vector of spin operators itself due to the spin commutation relations. The remaining symmetric part of $\boldsymbol{T}$ can, because $\text{Tr}(\boldsymbol{T}) = S(S+1) = 2$ is constant, be captured by five independent operators that can conveniently be organized into the vector

$$\boldsymbol{Q} := \begin{pmatrix} (S^x)^2 - (S^y)^2 \\ \frac{1}{\sqrt{3}}\left[ 2(S^z)^2 - S(S+1) \right] \\ S^x S^y + S^y S^x \\ S^y S^z + S^z S^y \\ S^z S^x + S^x S^z \end{pmatrix} \tag{2}$$

of *quadrupolar operators*.

Typically, spin-1 single-particle states have both a non-zero magnetic and a non-zero quadrupolar moment. However, when referring to a *quadrupolar state*, what we mean is a state that is solemnly described by $\boldsymbol{Q}$. In particular, we require that its magnetic dipole moment $m$ is zero. An example of a quadrupolar state is the $|0\rangle$ state in the $S^z$-basis. As can be checked directly: $m = 0$. Interestingly, even in the absence of magnetization, the $|0\rangle$ state does break spin-rotation symmetry because $\langle (S^z)^2 \rangle = 0$, whereas the fluctuations in x and y-direction are non-zero: $\langle (S^x)^2 \rangle = 1 = \langle (S^y)^2 \rangle$. Classically, this state can be thought of as a dipole moment that fluctuates in the x-y plane in such a way that it is zero on average. We will picture it by a disc portraying the plane of fluctuations (see Fig. 2). A normal vector to the plane of fluctuations is called a *director* ($\pm\boldsymbol{e}_z$ in the case of $|0\rangle$). Any quadrupolar single-particle state is fully described by the orientation of its directors.

A convenient on-site basis for investigating quadrupolar moments is the time-reversal invariant basis:

$$|x\rangle = \frac{i}{\sqrt{2}}(|\uparrow\rangle - |\downarrow\rangle), \; |y\rangle = \frac{1}{\sqrt{2}}(|\uparrow\rangle + |\downarrow\rangle), \; |z\rangle = -i|0\rangle, \tag{3}$$

where $\{|\uparrow\rangle, |0\rangle, |\downarrow\rangle\}$ is the standard $S^z$-eigenbasis. Invariance of the above vectors under the time-reversal operator $\mathscr{T}$, which is anti-unitary, follows immediately from the fact that $\mathscr{T}$ interchanges $|\uparrow\rangle$ and $|\downarrow\rangle$, and adds a sign to $|0\rangle$: i.e $\mathscr{T}|\uparrow\rangle = |\downarrow\rangle$, $\mathscr{T}|\downarrow\rangle = |\uparrow\rangle$, and $\mathscr{T}|0\rangle = -|0\rangle$.

Any real linear combination of the above basis states $\sum_{\alpha=x,y,z} u_\alpha|\alpha\rangle$ is also a quadrupolar state, with a director pointing along $\boldsymbol{u}$. More generally, an arbitrary spin-1 single-particle state can be expanded in the basis of (3) as $|\boldsymbol{d}\rangle = \sum_{\alpha=x,y,z} d_\alpha|\alpha\rangle$, where $\boldsymbol{d}$ has real and imaginary parts $\boldsymbol{d} = \boldsymbol{u} + i\boldsymbol{v}$. In terms of $\boldsymbol{u}$ and $\boldsymbol{v}$, the magnetic moment of $|\boldsymbol{d}\rangle$ is given by $\boldsymbol{S} = 2\boldsymbol{u} \times \boldsymbol{v}$. We can normalize the state and use global phase invariance to have $\boldsymbol{u}$ and $\boldsymbol{v}$ satisfy $u^2 + v^2 = 1$ and $\boldsymbol{u} \cdot \boldsymbol{v} = 0$. Assuming the last two equations hold, then so does the following: fully magnetized $m = 1$ states are precisely those for which $u = v$, whereas non-magnetic $m = 0$ states correspond to $(u, v) = (0, 1)$ or $(1, 0)$. The latter case ($m = 0$) describes purely quadrupolar states with a director $\boldsymbol{d}$ pointing along the direction of whichever of $\boldsymbol{u}$ or $\boldsymbol{v}$ is non-zero. Whenever $u$ and $v$ are both non-zero but *not* of equal magnitude, and consequently $0 < m < 1$, the state is of mixed character: neither fully magnetic nor purely quadrupolar.

Because the basis of (3) is invariant under the anti-unitary time-reversal operator $\mathscr{T}$, we can immediately conclude that the time-reversal invariant single-particle states are precisely those for which the coefficients in the above basis do not change under $\mathscr{T}$, up to a global phase. Since $\mathscr{T}$ is anti-unitary, this means that the time-reversal invariant states are precisely those for which $\boldsymbol{d}$ is either completely real ($v = 0$), or purely imaginary ($u = 0$); i.e. $|\boldsymbol{d}\rangle$ is a quadrupolar state. (This is most obvious in the above gauge $\boldsymbol{u} \cdot \boldsymbol{v} = 0$, where $u$ and $v$ both being non-zero results in $\boldsymbol{S}$ being flipped under $\mathscr{T}$.) In other words, following Andreev's definition [3] of a spin-nematic state as a state that breaks spin-rotational symmetry but preserves time-reversal symmetry, we conclude that, for a single spin-1 particle, the notions of quadrupolar and spin-nematic are equivalent.

Comparing to nematic order in liquid crystals, which is the alignment of rod-like particles in a liquid, we observe that quadrupolar states have the exact same symmetry properties as rods. Indeed: two directors $\boldsymbol{d}$ and $-\boldsymbol{d}$ correspond to the same quantum state (they differ by an overall phase). Contrast this to magnetic states, where the magnetic moment is described by the magnetic dipole vector, which is certainly not equal to minus itself. More explicitly, the magnetic the $|\uparrow\rangle$ and the quadrupolar $|0\rangle$ states are invariant (the first up to a phase) under rotations about the $z$-axis, but, in addition, the $|0\rangle$ state is (up to a minus sign) also invariant under a $\pi$-rotation about any axis that lies in the $x - y$ plane. Finally, we speak of nematic *order* when neighboring directors order in space, similar to the spatial ordering of rods in liquid crystals.

## 3.2 High-symmetry points

It shall not come as a surprise to the reader that the biquadratic term $(\boldsymbol{S}_i \cdot \boldsymbol{S}_j)^2$ in the Hamiltonian (1), which involves on-site products of spin operators, is related to quadrupolar order. However, part of the biquadratic term also includes the ordinary spin matrices (the antisymmetric part of $\boldsymbol{T}$ defined above), which are more naturally absorbed into the bilinear term. To better separate the two competing magnetic and quadrupolar interactions, let us rewrite the Hamiltonian in terms of $\boldsymbol{S}$ and $\boldsymbol{Q}$. Doing so yields

$$H = \sum_{\langle i,j \rangle} J_S(\theta)\boldsymbol{S}_i \cdot \boldsymbol{S}_j + J_Q(\theta)\boldsymbol{Q}_i \cdot \boldsymbol{Q}_j \tag{4}$$

up to an irrelevant $\theta$-dependent constant, with the spin and quadrupolar coupling constants given by $J_S(\theta) = \cos(\theta) - \sin(\theta)/2$ and $J_Q(\theta) = \sin(\theta)/2$.

We should remark that quadrupolar order is formally captured by the symmetric part of $\boldsymbol{T}$, and the set of operators in (2) is just one of many possible sets of operators that can be chosen to describe quadrupolar order. However, using the set of operators in (2) does unveil the at first sight hidden points of higher symmetry that the BBH Hamiltonian possesses. When expressing all three spin matrices and all five quadrupolar operators given by (2) in terms of the time-reversal invariant basis of (3): we observe that the spin matrices become equal to the three imaginary Gell-Mann matrices, whereas the above five quadrupolar operators equal the remaining five real Gell-Mann matrices. Recall that the Gell-Mann matrices are the generators of SU(3). Consequently, whenever the spin coupling constant equals the quadrupolar coupling constant, i.e. $J_S(\theta) = J_Q(\theta) = J$, the Hamiltonian reduces to an equal-weighted product over all Gell-Mann matrices: $H = J \sum_{\langle i,j \rangle} \boldsymbol{\lambda}_i \cdot \boldsymbol{\lambda}_j$, which is manifestly SU(3) invariant.[4] This occurs for $\theta = \pi/4$ and $\theta = 5\pi/4$. Additionally, when $J_S(\theta) = -J_Q(\theta) = J$, which happens for $\theta = \pi/2$ and $\theta = 3\pi/2$, all matrices again have equal weight, except that the imaginary matrices come with a relative extra minus sign. Since the square lattice is bipartite, we can compensate for this extra sign by taking the antifundamental representation $\bar{\boldsymbol{\lambda}} = -\boldsymbol{\lambda}^*$ of SU(3) on one sublattice. The Hamiltonian then takes the form $H = J \sum_{\langle i,j \rangle} \boldsymbol{\lambda}_i \cdot \bar{\boldsymbol{\lambda}}_j$ (where $i$ is in the A-sublattice, and $j$ in the B-sublattice), which is also SU(3) symmetric.

At the SU(3)-symmetric points, there is a larger set of operators commuting with the Hamiltonian, which means that the ground state degeneracy increases. Moreover, because the SU(3)-symmetric points are exactly the points at which the coupling constants are equal in magnitude ($|J_S| = |J_Q|$), the regions in between are precisely those for which one of the two coupling constants dominates the other. Hence, we can naively expect to have four different phases separated by the SU(3)-symmetric points, with magnetic or quadrupolar order depending on which of the two coupling constants is larger in magnitude.

## 4 Previous studies

In 1988 Papanicolaou [4] conducted a product-state analysis of the spin-1 BBH model, a concise summary of which can be found in [18]. Because the square lattice is bipartite, finding the product ground state reduces to a two-body problem (one product state per sublattice). Papanicolaou's results agree with our analysis above; he found four different phases separated by the SU(3) points: the familiar ferromagnetic (FM) and antiferromagnetic (AFM) phases, and in between a ferroquadrupolar (FQ) phase wherein neighboring sites have aligned directors, and a phase he called *semi-ordered* (see Fig. 3). The latter has a degenerate product ground state: minimizing the two-particle energy leads to the condition that one sublattice state has to be quadrupolar, whereas the other sublattice state can be either quadrupolar with a director perpendicular to that of the first sublattice, or magnetic with a magnetic moment aligned with the director of the first sublattice, or a combination of the two.

As an interesting side note, let us mention that exactly at each SU(3) point both product ground states left and right of the SU(3) point in question are simultaneous ground states of the system, and can be rotated into one another through some element of SU(3).

The product ground state degeneracy in the top right octant of the phase diagram means that we can expect quantum fluctuations to play an important role there. Moreover, the types

---

[4]Similar to SU(2) invariance of $\boldsymbol{S}_i \cdot \boldsymbol{S}_j$, where $\boldsymbol{S}$ is the vector of (a half times) the Pauli matrices (the generators of the Lie algebra of SU(2)), the inner product $\boldsymbol{\lambda}_i \cdot \boldsymbol{\lambda}_j$ over the generators of the Lie algebra of SU(3) is invariant under SU(3) rotations. Note that SU(2) and SU(3) act on their respective Lie algebras through the adjoint representation.

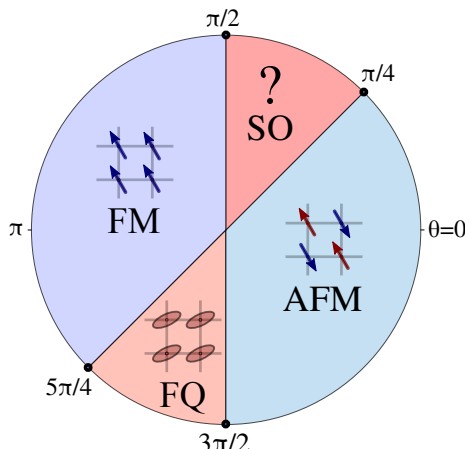

Figure 3: The product ground state phase diagram according to Papanicolaou [4]. In counterclockwise order starting at $\theta = 0$ we have the antiferromagnetic (AFM), semi-ordered (SO), ferromagnetic (FM) and ferroquadrupolar (FQ) phases. The SU(3)-symmetric points are marked with black dots.

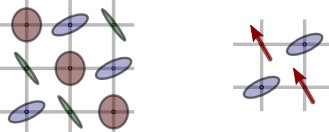

Figure 4: Different types of product ground states in semi-ordered phase ($\pi/4 \leq \theta \leq \pi/2$). Left: three-sublattice order with neighboring directors being perpendicular (antiferroquadrupolar 'AFQ3' state—discs with differing colors are quadrupolar states viewed at different angles); right: two-sublattice order with directors aligning parallel to neighboring magnetic moments (half nematic half magnetic '$m = 1/2$' state). Note that the color coding is only meant to highlight which particles are in the same quantum state.

of ground states found open up a lot of possibilities for interesting types of order;[5] for example: the fact that neighboring directors can be perpendicular in three different ways (e.g., pointing in x, y and z direction, respectively), allows for the possibility of a three-sublattice ground state (Fig. 4), which is surprising for a model with nearest-neighbor interactions on a bipartite lattice.

The energy per site for the product ground state is plotted in Fig. 5: it shows clear kinks at the SU(3)-symmetric points, suggesting that the phase transitions are all of first order.

The lower half $-\pi \leq \theta \leq 0$ of the phase diagram is free of the sign problem and has been studied using quantum Monte Carlo in 2002 by Harada and Kawashima [17], who found that the actual ground state phase diagram agrees with the product ground state phase diagram. Moreover, they observed jumps in the quadrupolar order parameter at the SU(3) points $\theta = 5\pi/4$ and $3\pi/2$, which agrees with the product state analysis and confirms that the phase transitions from the antiferromagnetic to the ferroquadrupolar and from the ferroquadrupolar to the ferromagnetic phases are both of first order.

---

[5]E.g., any three-color node coloring of the square lattice with neighboring nodes having different colors corresponds to a ground state, where each of the three colors stands for a quadrupole aligned in one of three mutually perpendicular directions; or, there are also fancier orderings, such as a two-sublattice ground state with $z$-aligned directors on the A-sublattice, and on the B-sublattice (possibly different) linear combinations of a $z$-magnetized state and any quadrupole with a director that lies in the x-y plane.

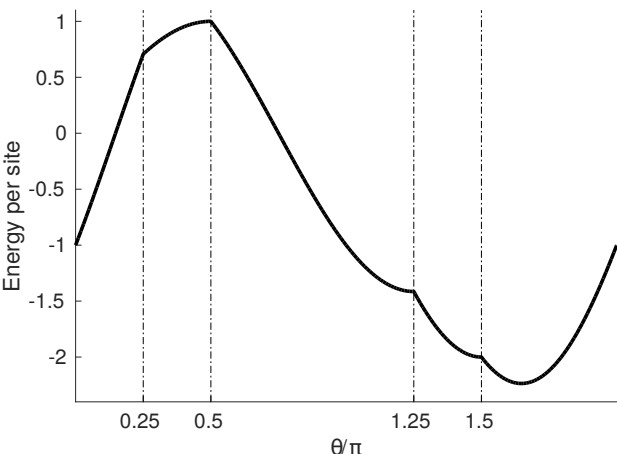

Figure 5: The product ground state energy. The kinks in the energy per site suggest that the phase transitions at the SU(3)-symmetric points are of first order.

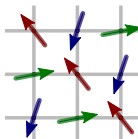

Figure 6: Magnetic order in the 120° magnetically ordered phase: the spins align in a plane and orient themselves at relative angles of 120°.

The Monte Carlo study was followed up by an exact-diagonalization and linear-flavor-wave study at the SU(3)-symmetric point $\theta = \pi/4$ by Tóth et al. [5] in 2010, who concluded that the ground state is actually a three-sublattice state that is favored over the two-sublattice state because it has a lower zero-point energy (the energy of the wave-excitations on top of the classical ground state averaged over the Brillouin-zone). The three-sublattice order of the ground state at the SU(3) point was confirmed by DMRG and iPEPS simulations by Bauer et al. [6].

A further study by Tóth et al. [18] in 2012 based on exact diagonalization and linear-flavor-wave theory examined the question whether the above-mentioned three-sublattice phase extends beyond the SU(3) point $\theta = \pi/4$. They found that, in the semi-ordered phase, the ground state is the three-sublattice *antiferroquadrupolar* (AFQ3) state with perpendicular directors on neighboring sites (the analogous product state is depicted in Fig 4: left). Interestingly, Tóth et al. also found that the three-sublattice pattern extends into the antiferromagnetic regime, where, for $0.2\pi \lesssim \theta < \pi/4$, instead of ordinary antiferromagnetism, a 120° magnetic order emerges (Fig. 6). This result was thereafter confirmed by series expansion [19].

# 5 iPEPS results

## 5.1 Overview

To get a first rough picture of the ground state phase diagram, and also as a consistency check, we initialize $D = 1, 2, 3, \ldots, 6$ simple update simulations for $\theta$ at 40 evenly spaced points covering all of $[0, 2\pi]$ using 2x2 and 3x3 unit cells (Fig. 7; only $D = 1, 2$ and 6 shown). The

top plot of Fig. 7 shows the energy per site, which is seen to decrease as the bond dimension increases, except in the FM region $\pi/2 < \theta < 5\pi/4$, where the ground state can be represented by a product state (i.e. $D = 1$). Moreover, we observe a transition from a two-sublattice to a three-sublattice ground state around $\theta \approx 0.2\pi$, with the $0.2\pi$ three-sublattice simulations showing 120° magnetic order observed in the AFM3 phase.

The bottom plot of Fig. 7 shows the magnetization per site (for $D = 6$ simulations), which is of order one in the FM and AFM phases, and zero in the quadrupolar phases, as expected. Contrary to magnetic order, quadrupolar order cannot be captured by a single order parameter, because it is described by a matrix: the traceless symmetric part of $S^\alpha S^\beta$, denoted by $Q^{\alpha\beta}$.[6] The convention used for the quadrupolar operators in the vector $\boldsymbol{Q}$ given by Eq. (2) has a preferred choice of direction in spin-space, as can be seen from the first two components of $\boldsymbol{Q}$ which represent the diagonal part of $Q^{\alpha\beta}$. Using $\boldsymbol{Q}$ works well for nematic order along the $z$-direction; however, we do not control the direction of spontaneous symmetry breaking in our simulations. Therefore, when measuring quadrupolar order, we prefer to work with invariants of $Q^{\alpha\beta}$ such as its eigenvalues, or equivalently, the matrix invariants $I_Q = \mathrm{tr}(Q) = 0$, $II_Q = \frac{1}{2}\left(\mathrm{tr}(Q)^2 - \mathrm{tr}(Q^2)\right) = -\frac{1}{2}\mathrm{tr}(Q^2)$ and $III_Q = \det(Q)$ rather than the individual components of $\boldsymbol{Q}$. Fig. 7 shows the two non-zero matrix invariants $II_Q$ and $III_Q$, which are clearly larger in magnitude in the quadrupolar phases than they are in the magnetized phases.[7]. Moreover, the determinant $III_Q$ changes sign when going from a nematic to a magnetic state

The $D = 1, 2, \ldots, 6$ simple update simulations reproduce the AFM, AFQ3, FM and FQ phases separated by the SU(3) points, and hint at the existence of the AFM3 phase[8] found by Tóth et al. [18] in between the AFM and AFQ3 phases. Because the lower half $-\pi < \theta < 0$ of the phase diagram has already been established by quantum Monte Carlo [17], and in the top left quadrant $\pi/2 < \theta < \pi$ we only found ferromagnetic translationally invariant product states that have the same energy independent of $D$ or unit cell size, the only section of the phase diagram that remains to be thoroughly investigated is the top right quadrant $0 < \theta < \pi/2$. In order to obtain more accurate data on this interesting region, we will use finer-spaced higher-$D$ full update simulations rather than the less accurate simple update simulations shown in Fig. 7.

Part of the top right quadrant we have already tackled in our recent paper [20], where we attempted to determine the extent of the AFM3 phase, and in the process stumbled upon the paramagnetic (extended) Haldane phase that appears in between the AFM and AFM3 phases. Next, we turn our attention to the quadrupolar part of the top right quadrant, and find yet another unexpected phase: the partially nematic partially magnetic '$m = 1/2$' phase that we shall discuss in detail below.

Combining all of our iPEPS results, we arrive at the phase diagram shown in Fig. 8. Note that the Haldane, AFM3 and $m = 1/2$ phases are not visible in the simple update plots in Fig. 7 because the $\theta$-grid used is too coarse, and, in case of the Haldane phase, the bond dimension $D$ is too low.

In the following section, we will go through all phases in Fig. 8 in anti-clockwise order and discuss their properties, and, where necessary, elaborate on our findings. In the section thereafter, we will discuss the nature of all occurring phase transitions—also shown in Fig. 8. The analysis presented below involves simulations with a bond dimension as high as $D = 10$ ($D = 16$ for simulations in the Haldane phase), followed by a $D \to \infty$ extrapolation where necessary. As a consequence, the expectation values may differ slightly from those shown in Fig. 7 which are only meant to give a rough overview of the type of order that can be expected.

---

[6]I.e. the matrix $Q^{\alpha\beta} = S^\alpha S^\beta + S^\beta S^\alpha - \frac{2}{3}S(S+1)\delta^{\alpha\beta}$. (The last term makes it traceless.)

[7]For a typical spectrum of $Q$, see Fig. 15 in the appendix.

[8]The iPEPS simulations at $\theta = 0.2\pi$ show that the state with the 3x3 unit cell is competitive in energy, and all of the corresponding 3x3-unit-cell simulations display the AFM3 magnetization pattern.

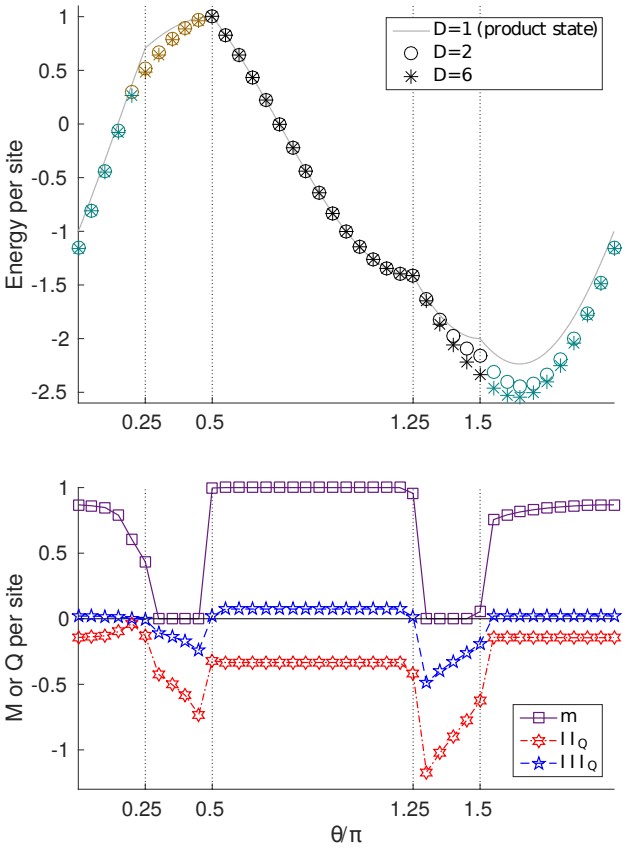

Figure 7: *Top*: energy per site for $D = 1, 2, 6$ simple update simulations using 2x2 and 3x3 unit cells; for each value of $\theta$ only the lowest (in energy) of the two is shown. The black markers correspond to a 1x1 unit cell state (energies for 2x2 and 3x3 simulations coincide). *Bottom*: magnetization per site $m$ and the $II_Q$ and $III_Q$ tensor invariants of the $Q$-matrix per site for the $D = 6$ simulations. $I_Q$ is identically zero (not shown).

## 5.2 Description of all phases

### 5.2.1 AFM

The familiar antiferromagnetic phase can be described by a 2x2 two-sublattice unit cell, with spins pointing in opposite directions on each sublattice. The magnetization per site varies throughout the phase from $m \approx 0.5 - 0.6$ close to the FQ phase (Fig. 14) to $m \approx 0.8$ around $\theta = 0$, after which it monotonically decreases to zero or almost zero when approaching the Haldane phase [20]. States in the AFM phase are U(1)-spin-rotation symmetric around the axis of magnetization.

### 5.2.2 Haldane

As described in our recent paper [20], we set about to nail down the precise value of $\theta_c$ for which the proposed antiferromagnetic to 120° magnetically ordered phase transition was supposed to occur. While doing so, we discovered that, as we increased $\theta$ within the AFM phase, the magnetization vanished before the transition to the AFM3 phase, hinting at the existence of an intermediate phase in between the AFM and AFM3 phases. Further investigation showed that the lowest energy per site was obtained by a 1x1 unit cell iPEPS which converged to a paramagnetic state that preserves spin-rotation (hence also time-reversal) and lattice translation

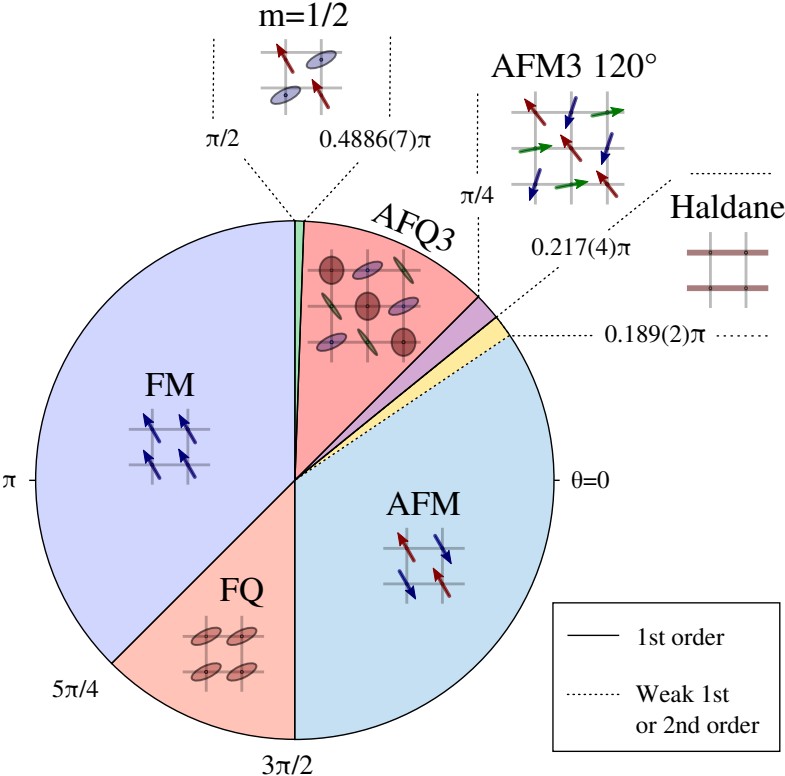

Figure 8: The phase diagram according to our iPEPS data. All phase transitions are of first order (solid lines), except possibly the AFM to Haldane phase transition, which is a weak first or second order phase transition (dotted line).

symmetry, but breaks lattice rotation symmetry—in the sense that there is an energy difference between the x and y-bonds—reminiscent of a system of coupled one-dimensional chains. This led us to investigate the anisotropic BBH model, in which we introduced an additional coupling parameter $(0 \leq J_y \leq 1)$ for the y-bonds only that allowed us to interpolate between the isotropic two-dimensional system and the one-dimensional system of decoupled chains. We then identified the intermediate paramagnetic phase as the Haldane phase, by showing that it can be adiabatically connected to the one-dimensional Haldane phase of decoupled spin-1 BBH chains by sending $J_y$ to zero.

As a side note, we should remark that the Haldane phase we find is not related to the two-dimensional AKLT state. The latter can be obtained by taking four spin-1/2 particles per site—each forming a singlet bond with a spin-1/2 particle on a neighboring site—and projecting onto the on-site spin-2 subspace, which results a state that has a spin-2 particle on each lattice site. Rather, the analogous picture for the state we find is that of (decoupled) one-dimensional AKLT chains, each having two spin-1/2 particles per site that form singlet bonds only in one direction before projecting onto the on-site spin-1 subspace.

The location of the AFM to Haldane phase transition (at $\theta = 0.189(2)\pi$) we have determined by investigating for what value of $\theta$ the magnetization per site in the AFM phase vanishes. The phase transition from the Haldane to the AFM3 phase (at $\theta = 0.217(4)\pi$) follows from an energy per site comparison of simulations in both phases, in the same spirit as the AFQ3 to $m = 1/2$ phase transition discussed below.

### 5.2.3   AFM3

The three-sublattice 120° magnetically ordered phase partially breaks translational symmetry in a 3x3 unit cell pattern, and has the spins on the three sublattices aligned in a plane such that they are at relative angles of 120° (see Fig. 6). The magnetization per site varies from $m \approx 0.4$ at $\theta = 0.22\pi$ [20] to $m \approx 0.1 - 0.3$ near $\theta = \pi/4$ (Fig. 10). AFM3 states have no residual spin-rotation symmetry.

### 5.2.4   AFQ3

Similar to the AFM3 phase, the three-sublattice antiferroquadrupolar phase is also described by a 3x3 unit cell, but now the magnetization per site is near zero at $\pi/4$ (Fig. 10)[9], and exactly zero from about $\theta \approx 0.27\pi$ up to the $m = 1/2$ phase (see Fig. 11 for $\theta = 0.487\pi$). At the product state level, in addition to being time-reversal symmetric, the AFQ3 state has a residual spin-rotation symmetry given by $\pi$-rotations around the (mutually perpendicular) axes of nematic polarization.

### 5.2.5   $m = 1/2$

Next, let us focus on the extent of the three-sublattice AFQ3 phase in the semi-ordered region $\pi/4 < \theta < \pi/2$. Recall that, at the product state level, there is a multitude of distinct types of ground states, two of which are depicted in Fig. 4. Tóth et al. [18] predicted that the three-sublattice (AFQ3) state is the ground state for $\pi/4 < \theta < \pi/2$. However, their product state analysis in this same region shows that the addition of an infinitesimal external magnetic field favors the two-sublattice half nematic half magnetic $m = 1/2$ state as the ground state. Augmented by linear-flavor-wave theory, they continue to show that the AFQ3 phase extends into the small but non-zero magnetic field $h > 0$ region for $\pi/4 < \theta < \pi/2$, but that, as $\theta$ approaches $\pi/2$ from below, the phase boundary between the $m = 1/2$ and AFQ3 phases moves down to $h \to 0$ as $\theta \to \pi/2$. Since their exact diagonalization results for the AFQ3 phase with non-zero magnetic field do not extrapolate well to infinite system size, and linear-flavor-wave theory is semi-classical, it is not clear what happens close to $\theta = \pi/2$.

We have thoroughly investigated the parameter range of $0.48\pi < \theta < \pi/2$ using iPEPS. To get an idea of what types of states to expect, we initialized simulations using eleven different types of unit cells (to also allow for possibilities such as stripe order, as well as the more conventional types of order that can be represented by unit cells up to size 4x4[10]) and evolved them in imaginary time using the simple update algorithm up to bond dimension $D = 6$. We then picked the minimal unit cell for each type of state obtained (e.g. the 4x2 state displayed the same pattern as the 2x2 state, so we discarded the former), and evolved those in imaginary time using the full update algorithm up to $D = 8$. This left two competitive states: the three-sublattice AFQ3 state described by a 3x3 unit cell iPEPS, and the two-sublattice $m = 1/2$ state described by a 2x2 unit cell iPEPS. Interestingly, the $m = 1/2$ state turned out to have a lower energy than the AFQ3 state for $0.49\pi \lesssim \theta < \pi/2$. Finally, we ramped up the bond dimension to $D = 10$ to nail down the precise location of the phase transition between the AFQ3 and $m = 1/2$ phases.

As can be seen in Fig. 9, for $\theta = 0.487\pi$, the AFQ3 simulation is lower in energy than the $m = 1/2$ simulation, whereas for $\theta = 0.490\pi$ the opposite is true. Taking the error bars

---

[9]Possibly, the magnetization is not exactly zero at $\pi/4$ and $5\pi/4$ because of the higher SU(3) symmetry that allows magnetic and quadrupolar states to be rotated into each other at the cost of no energy, which weakens the hysteresis effect and prevents the quadrupolar simulation from remaining completely nematic at the phase transition.

[10]Specifically, we have initialized simple update simulations using 2x2, 3x2, 3x3, 4x2, 4x3, 4x4, 5x2, 5x3, 6x2, 7x2, and 8x2 sized unit cells.

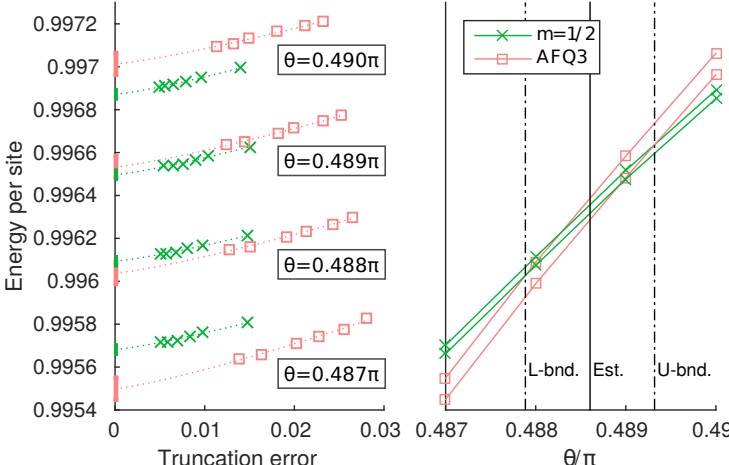

Figure 9: The phase transition between the AFQ3 and $m = 1/2$ phases. *Left:* energy per site for bond dimensions $D = 5, 6, \ldots, 10$ as a function of truncation error, which we extrapolate to zero. *Right:* extrapolated energy per site as a function of $\theta$. We estimate the phase transition to occur at the point where the energies per site intersect, i.e. at $\theta = 0.4886(7)\pi$.

from the $D \to \infty$ extrapolation into account, we can conclude that—in contrast to previous predictions—the AFQ3 phase does *not* persist all the way up to the SU(3)-symmetric point $\pi/2$, but remains stable only up to $\theta = 0.4886(7)\pi$, after which the system transitions into the $m = 1/2$ phase.

In case the reader is wondering why the $m = 1/2$ phase occupies such a small portion of the phase diagram, it is insightful to remark that the size in $\theta$-space is not a physical quantity, as $[\cos(\theta), \sin(\theta)]$ is but one of the many possible parameterizations of the coupling constants of the Hamiltonian in Eq. (1). Additionally, there is a reason for the $m = 1/2$ phase to only appear this close to $\pi/2$. The actual ground state is not a product state, and on the quadrupolar sublattice a small magnetic moment parallel to that of the magnetized sublattice can be detected. This is only energetically favorable when the Heisenberg coupling parameter $J_S(\theta)$ in the Hamiltonian expressed in terms of $\boldsymbol{S}$ and $\boldsymbol{Q}$ (see Eq. (4)) becomes of similar magnitude as the quadrupolar coupling parameter $J_Q(\theta)$, which only happens around $0.49\pi$ where $J_S(\theta)$ decreases rapidly from 0 towards $-1/2$.

As its name suggests, the magnetization per site of $m = 1/2$ states is exactly $1/2$ throughout the entire phase (e.g., see Figs. 11 and 12 for $\theta$ close to the transition points). It has a residual U(1)-spin-rotation symmetry around the axis of magnetization. As $\theta$ approaches $\pi/4$, the $m = 1/2$ state gradually turns into a product state (see Section 5.3.5).

### 5.2.6 FM

The familiar ferromagnetic phase is described by a translationally invariant product state represented by a 1x1 unit cell. It has a residual U(1)-spin-rotation symmetry around the axis of magnetization, and a magnetization per site of exactly $m = 1$ (see Figs. 7, 12 and 13).

### 5.2.7 FQ

Finally, the ferroquadrupolar phase is also translationally invariant, and can be represented by a 1x1 unit cell. Its magnetization is zero or very close to zero throughout the phase (see Figs. 13 and 14 for the transition points, and Footnote 9). At the product state level, it is

symmetric under time-reversal, residual U(1)-spin-rotations around the axis of nematic order, and $\pi$-rotations around any axis perpendicular to the axis of nematic order. Close to the FM phase, states in the FQ phase are product states, but, as we approach the AFM phase, the FQ states become gradually more and more entangled (see Sections 5.3.6 and 5.3.7).

## 5.3 Nature of the phase transitions

In the following section, we will discuss the nature of the phase transitions in the phase diagram shown in Fig. 8, including the transitions at $0.189(2)\pi$ and $0.217(4)\pi$ that have already been discussed in [20] (a brief summary of which will be presented below).

From the previous section, we gather that all neighboring phases break different symmetries, which suggests that all phase transitions are either first order, or unconventional second order transitions (see e.g. [60, 61]). To support this hypothesis—and where possible distinguish between the two options—we will next look for kinks in the energy per site or jumps in typical order parameters such as the magnetization or the Q-matrix invariants per site due to varying $\theta$.

### 5.3.1 AFM to Haldane: $\theta = 0.189(2)\pi$

At $\theta = 0.189(2)\pi$, we have a transition between the AFM and Haldane phases. In the AFM phase, we observed that the magnetization per site goes to zero when approaching the Haldane phase suggesting a second order phase transition, which is unconventional considering the fact that both states break different lattice translation and rotation, as well as different spin-rotation symmetries. Moreover, we did not find clear hysteresis behavior, which supports the claim that this transition is second order. However, due to the error bars in the magnetization close to the transition, we were not able to say with certainty whether the phase transition is a second or a weak first order transition; all we can say for sure is that this is not a clear first order transition as no jump in magnetization or kink in the energy were observed.

### 5.3.2 Haldane to AFM3: $\theta = 0.217(4)\pi$

The phase transition between the Haldane and the three-sublattice AFM3 phase displays a clear jump in magnetization, which goes from zero (Haldane) straight to 0.4 (AFM3) at the transition point. This implies that the transition is first order.

### 5.3.3 AFM3 to AFQ3: $\theta = \pi/4$

At $\theta = \pi/4$, we have a transition from the three-sublattice 120° magnetically ordered phase to the three-sublattice antiferroquadrupolar phase. Precisely at the symmetric point $\theta = \pi/4$, both the 120° magnetically ordered state and the antiferroquadrupolar state are ground states of the system. Hence, we can simulate both at the phase transition by initializing the simulations from deep within the 120° magnetically ordered and quadrupolar phases, respectively, and then moving towards the phase transition by initializing each simulation from the previous one. The resulting data at the critical point is shown in Fig. 10.

The left plot of Fig. 10 shows a subtle kink in the energy per site, which we observe for each fixed bond dimension simulation for $D = 2, 3, \ldots, 8$ (only $D = 4, 6, 8$ shown), supporting the occurrence of a first order transition. The right plot shows the magnetization per site exactly *at* the phase transition (where we increased $D$ up to 10). Despite the fact that the magnetization data points do not fit on a perfect straight line, a rough extrapolation of $D \to \infty$ shows that the magnetic AFM3 state at the transition has a magnetization of at least 0.1, whereas the quadrupolar AFQ3 state has a magnetization of around zero[9], which agrees with the above observation that this is a first order transition.

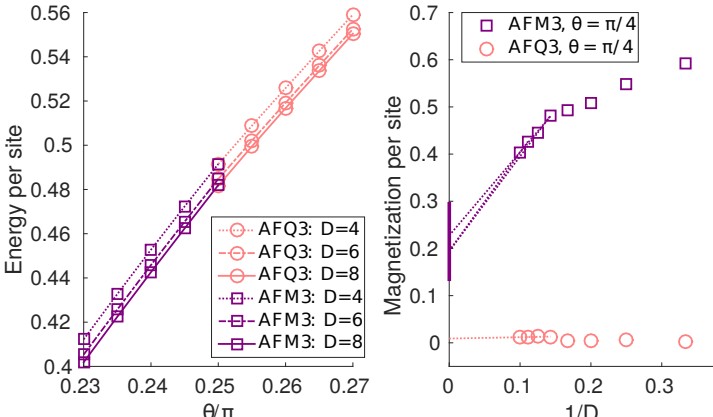

Figure 10: Energy per site (left) and magnetization per site (right) for the 120° magnetically ordered (AFM3) and the antiferroquadrupolar (AFQ3) states near the phase transition at $\theta = \pi/4$. By carefully comparing the slopes of both graphs, we observe a very subtle kink in the energy as a function of $\theta$ for each fixed $D$ (only $D = 4, 6, 8$ plotted; more clutters the figure), and the extrapolated magnetization (plotted for $D = 3, 4, \ldots, 10$) jumps from $m > 0.1$ (AFM3) to $m \approx 0$ (AFQ3), indicating a first order transition.

Because the magnetization does not extrapolate very nicely to $D \to \infty$, we have also investigated the behavior of the eigenvalues of the quadrupole matrix (Appendix: Fig. 15 right), where a jump in spectrum can be observed. Additionally, we have also observed the occurrence of hysteresis (Appendix: Fig. 15 left). All of the above taken together, combined with the fact that, at the product state level, the two phases break different symmetries, lead us to conclude that this must be a first order phase transition.

### 5.3.4 AFQ3 to $m = 1/2$: $\theta = 0.4886(7)\pi$

Using the same simulations as the ones shown in Fig. 9 (but now only for $\theta = 0.487\pi$ and $\theta = 0.490\pi$ to prevent the figure from getting too cluttered), we see that the magnetization on both sides of the phase transition for the $m = 1/2$ states is exactly one-half, whereas it is exactly zero for the antiferroquadrupolar states (Fig. 11). Thus, at the transition from AFQ3 to $m = 1/2$ at $\theta = 0.4886(7)\pi$, the magnetization jumps from zero to one-half, which means that the transition is of first order. This notion is confirmed by the right plot of Fig. 9, which shows that, at the intersection, the energy per site graphs for the AFQ3 and $m = 1/2$ phases have different slopes, indicating a kink in the energy per site at the transition.

### 5.3.5 $m = 1/2$ to FM: $\theta = \pi/2$

In this case, the transition is between two magnetized phases: the half-magnetized $m = 1/2$ phase and the ferromagnetic phase. States in the ferromagnetic phase are product states, which in our simulations can be seen by the fact that the energy does not improve as the bond dimension increases. Hence, any fixed $D$ simulation will reproduce the exact ground state ($D = 2$ shown in Fig. 12).

As $\theta$ approaches $\pi/2$ from below, the $m = 1/2$ state also turns into a product state, which can be seen from the fact that the the energy per site graphs for different values of $D$ all converge to the same value at $\theta = \pi/2$ (Fig. 12 left). Hence, at the phase transition, no $D \to \infty$ extrapolation is necessary, as the $D = 2$ results already give the exact ground state.

The left plot of Fig. 12 displays a clear kink in the energy, demonstrating that this is a first

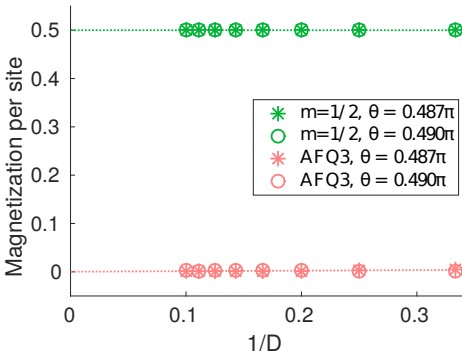

Figure 11: Magnetization per site for the half magnetized half nematic ($m = 1/2$) and the three-sublattice antiferroquadrupolar (AFQ3) states left and right of the phase transition at $\theta = 0.4886(7)\pi$ for $D = 3, 4, \ldots, 10$. The former clearly extrapolates to $1/2$, the latter to zero. Hence, the transition is of first order.

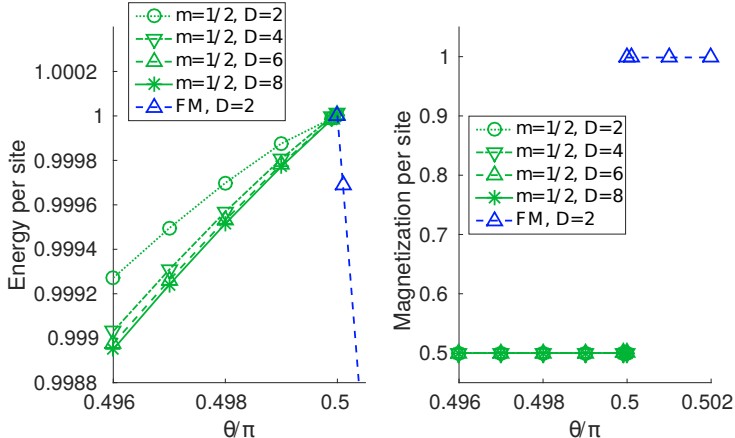

Figure 12: Energy per site (left) and magnetization per site (right) for the partially magnetized ($m = 1/2$) and ferromagnetic (FM) phases as a function of $\theta$. Because both states are product states at the critical point, no $D \to \infty$ extrapolation is needed. The kink in energy and jump in magnetization show that this is a clear first order transition.

order transition. Moreover, Fig. 12 right shows that the $m = 1/2$ phase has a magnetization per site of exactly $1/2$, whereas the ferromagnetic phase is fully magnetized, confirming that the transition is indeed of first order.

The remaining two critical points at $5\pi/4$ and $3\pi/2$ have previously been investigated using quantum Monte Carlo simulations [17]. Harada and Kawashima demonstrated that $(S^z)^2 - 2/3$, which they used as the quadrupolar order parameter, exhibits a jump at $\theta = 5\pi/4$ and at $\theta = 3\pi/2$. This indicates that both transitions are of first order; a conclusion that also follows from our simulations, which we present below for completeness. Because the jump in the quadrupole moment has already been established, we will focus on the energy and magnetization in the following, but remark that we also observe clear jumps in the spectrum of the $Q$-matrix.

### 5.3.6 FM to FQ: $\theta = 5\pi/4$

As before, we are dealing with product states on both sides of the phase transition. Thus, we can use any fixed $D$ simulations to investigate the nature of the ferromagnetic to ferro-

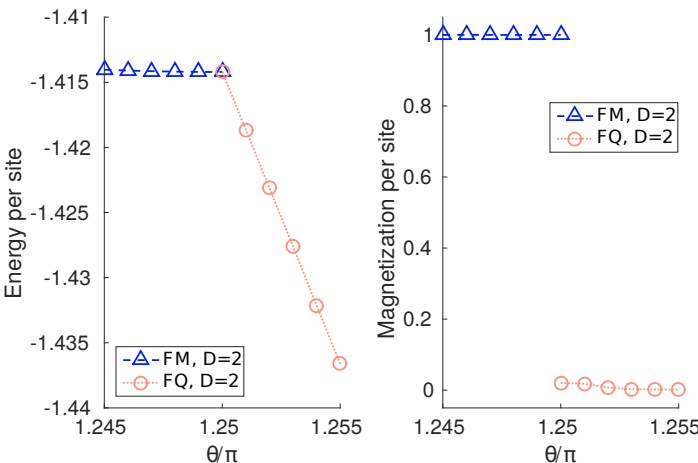

Figure 13: Energy per site (left) and magnetization per site (right) for the ferromagnetic (FM) and ferroquadrupolar (FQ) states as a function of $\theta$. No extrapolation is needed because both phases are represented by product states. The kink in energy and jump in magnetization show that this transition is first order.

quadrupolar phase transition ($D = 2$ shown). Both the kink in energy and the jump in magnetization displayed in Fig. 13 confirm that the phase transition at $\theta = 5\pi/4$ is indeed of first order.

### 5.3.7 FQ to AFM: $\theta = 3\pi/2$

When increasing $\theta$ from $5\pi/4$ to $3\pi/2$, the ground state becomes more and more entangled while remaining in the ferroquadrupolar phase. Upon reaching $3\pi/2$ the ground state is no longer a product state, and therefore we require higher $D$ simulations to investigate the phase transition at $3\pi/2$.

As shown by Fig. 14, there is a subtle but observable kink in the energy per site as a function of $\theta$. Additionally, the magnetization per site at the phase transition, which does not extrapolate very well to $D \to \infty$, does appear to saturate in between 0.5 and 0.6 in the AFM phase, whereas it extrapolates to zero in the FQ phase, confirming that this transition is also of first order.

## 6 Conclusion

We have studied the ground state phase diagram of the spin-1 BBH model on the square lattice by means of infinite projected entangled pair states (iPEPS). Using low bond dimension simple update simulations, we were able to reproduce all phases that had previously been predicted to occur [4,18] in this model. We then turned our attention to the challenging top right quadrant of the phase diagram: the part that has the largest product ground state degeneracy, is beyond the reach of Monte Carlo, and had up to now only been studied on small systems with exact diagonalization [18] or by means of semi-classical approaches [18,19].

In this interesting region, we found two new phases: the paramagnetic extended Haldane phase (discussed in our previous work [20]) that preserves O(3)-spin and lattice translational symmetry, but breaks lattice rotational symmetry and can be adiabatically connected to the one-dimensional Haldane phase; and the $m = 1/2$ phase that had been known to exist in the presence of an external magnetic field [18], but, in contrast to previous predictions, our

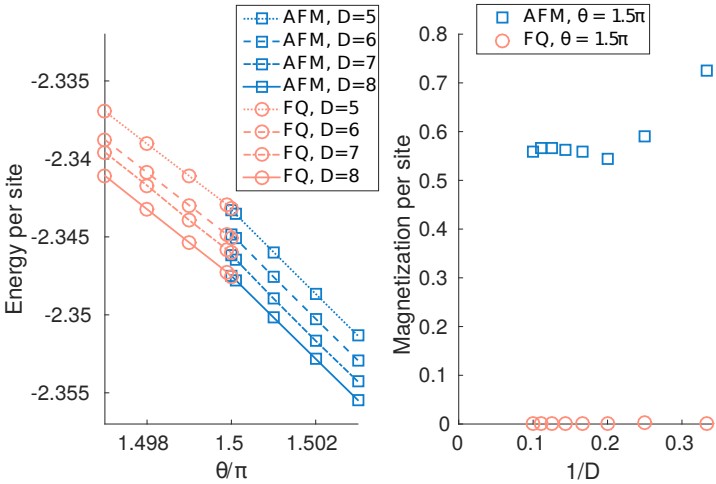

Figure 14: Energy per site as a function of $\theta$ for $D = 5, \ldots, 8$ (left) and magnetization per site exactly at the phase transition at $\theta = 3\pi/2$ for $D = 3, 4, \ldots, 10$ (right) for the ferroquadrupolar (FQ) and antiferromagnetic (AFM) phases. The energy per site shows an observable kink for each $D$ plotted. Moreover, the magnetization per site is zero in the FQ phase, whereas it saturates in between 0.5 and 0.6 in the AFM phase, showing that the transition is first order.

iPEPS data showed it also exists in a small $\theta$-window without external field. We concluded our analysis by investigating the nature of all phase transitions of the BBH model, mainly by looking at kinks in the energy and jumps in the magnetization per site as a function of $\theta$, while also taking symmetry considerations into account. We found clear or subtle kinks in the energy for all transitions but the AFM to Haldane transition. Similarly, the magnetization displayed slight to clear jumps except at the above-mentioned transition, leading us to conclude that all phase transitions are of first order, except possibly the AFM to Haldane transition, which we predict to be either of second or weak first order.

In addition to demonstrating that the spin-1 BBH model on the square lattice exhibits various exotic phases, such as several (partially) nematic phases ($m = 1/2$, FQ and AFQ3), three-sublattice ordering (AFM3 and AFQ3), and even a highly symmetric paramagnetic phase that only breaks lattice rotational symmetry (Haldane), which are interesting in their own right from a theoretical point of view, we hope to have paved the way for a future investigation of the BBH model on the numerically more challenging triangular lattice, possibly offering insight into the not-yet understood behavior of $NiGa_2S_4$ [11–13] and $Ba_3NiSb_2O_9$ [14–16]. Finally, our results show that iPEPS is a competitive method for analyzing strongly correlated spin systems, especially where quantum Monte Carlo suffers from the sign problem.

**Funding information**   This project has received funding from the European Research Council (ERC) under the European Union's Horizon 2020 research and innovation programme (grant agreement No 677061). This work is part of the Delta-ITP consortium, a program of the Netherlands Organization for Scientific Research (NWO) that is funded by the Dutch Ministry of Education, Culture and Science (OCW). This research was supported in part by Perimeter Institute for Theoretical Physics. Research at Perimeter Institute is supported by the Government of Canada through Industry Canada and by the Province of Ontario through the Ministry of Research and Innovation.

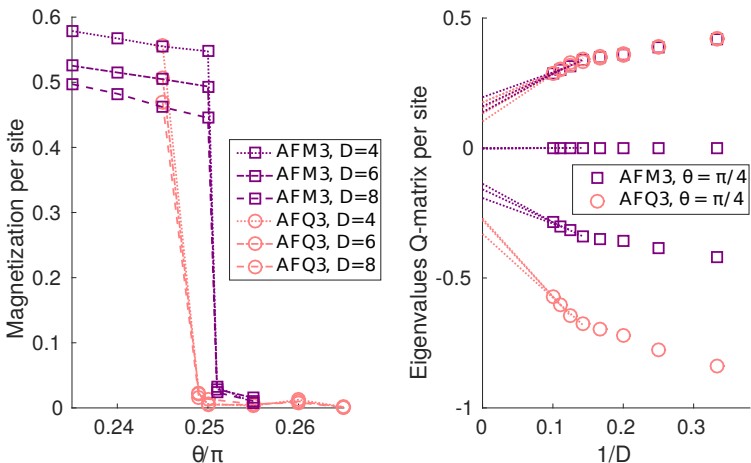

Figure 15: *Left:* magnetization per site for $D = 4, 6, 8$ as a function of the coupling parameter $\theta$ for the 120° magnetically ordered (AFM3) and antiferroquadrupolar (AFQ3) phases—hysteresis can be observed. *Right:* eigenvalues of the $Q$-matrix for the AFM3 and AFQ3 states at the phase transition at $\theta = \pi/4$ for $D = 3, 4, \ldots, 10$. Extrapolating $D \to \infty$ shows a jump in the spectrum of the $Q$-matrix.

# A    Additional data

The appendix contains additional data to support the main text. We have included plots that provide extra evidence for the occurrence of a first order transition at $\theta = \pi/4$. For the other transitions, the jump in magnetization or kink in the energy is clear enough to draw conclusions from. Additional plots can be provided upon request.

At $\theta = \pi/4$, hysteresis can be observed around the phase transition (Fig. 15 left): the quadrupolar phase specifically can be simulated at $\theta = 0.249\pi < \pi/4$ (where the ground state is in the magnetized phase) before jumping to a magnetized state at $\theta = 0.245\pi$.

Also, the eigenvalues of the $Q$-matrix (Fig. 15 right) differ in both phases: a rough extrapolation of $D \to \infty$ shows that in the antiferroquadrupolar phase we have eigenvalues of approximately 0.15 (twice) and $-0.3$ once, whereas in the 120° magnetically ordered phase we have 0.15, 0 and $-0.15$, indicating a jump in the spectrum, which leads us to conclude that the transition is of first order.

Additionally, from the spectrum of the $Q$-matrix we obtain some extra information about the quadrupolar phase: because $Q$ has two identical eigenvalues, the nematic order is uniaxial (as opposed to biaxial).

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
