# Peer review of "A tensor network study of the complete ground state phase diagram of the spin-1 bilinear-biquadratic Heisenberg model on the square lattice"

_SciPost Physics, doi:SciPost Phys. 3, 030 (2017)_

## Round 2 · Referee Report · Anonymous · 2017-8-24

Strengths
1) Interesting system
2) Useful to use the IPEPS method, the results appear to be reliable
3) New phases are identified
4) The paper is well organized
Weaknesses
1) Some lack of discussion of the findings (see report)
Report
The paper is interesting and I consider the numerical results reliable. In my opinion the most interesting aspect is that the authors identify a "Haldane" phase - a 2D phase claimed to be adiabatically connected to the ground state of the spin-1 chain (the conventional Haldane state). This is also where I think some more discussion is required. The Haldane phase in 1D is adiabatically connected to the AKLT state, which can be pictorially described as the S=1 spins on a chain being "split up" into S=1/2 objects forming on-site triplets. The S=1/2 objects on nearest-neighbor sites form singlets (before the projection to triplets on the sites - after the projection the bonds consist of mixed singlets and triplets, with no S=2 components). This kind of AKLT state also exists in 2D, but with S=2 spins. It is not clear how this kind of state should be understood in a system of S=1 spins (how would the valence bonds form? some kind of resonating bonds?). Naively, it appears that the nature of the state must be quite different and it should not be adiabatically connected to the Haldane chain. This naive thinking may very well be wrong but I think the authors have to discuss the nature of this state and, if possible, present some more results supporting their claims.
Requested changes
1) Discuss the nature of the Haldane phase as discussed above and present more results supporting the claims.

---

## Round 2 · Referee Report · Anonymous · 2017-8-24

Strengths
1) Complete the phase diagram of a relevant spin 1 model on the square lattice using relatively well established tensor network techniques.
2) It unveils a new phase
3) It analyses the nature of the observed phase transitions.
4) Complements and confirms the previous results
5) It performs a rigorous analysis of the systematic approximations involved in the numerical method
Weaknesses
1) Does not use the best available algorithms in a region where competing states belonging to different phases have very close energies ( they differ at most by 0.1% in Fig 8.).
2) It uses a lot of jargon rendering the paper fully accessible only to experts in the sector.
Report
The paper presents solid results obtained with iPEPS on the phase diagram of the spin 1 bilinear bi-quadratic Heisenberg model.
The model is parametrized in terms of an angle \theta so that the full phase diagram is contained in the 0 2\pi interval.
The paper tries to summarize the previous results on the topic.
It points out that the region of the phase diagram between \pi and 2 \pi is known from quantum Monte Carlo studies, while the region between 0 and \pi has been investigated either approximately Ref. [4] or on small systems with ED Ref. [5]. The special point where \theta = pi/4 had also already been studied in Ref [6]. Furthermore in Ref. [21] the authors have unveiled a region of parameters in which the model exhibits an Haldane phase.
The present work seems to settle the last piece of the puzzle by identifying a new intermediate nematic partially polarized phase close to pi/2.
The numerical analysis is sound and an accurate study as a function of the bond dimension, including an extrapolation is performed to ensure that the results are correct (see Fig. 8) (On this point though I have a main concern and I would appreciate further clarifications on the chosen algorithm).
For this reason I believe it constitutes an important and sound contribution . Once my main concern has been addressed I recommend the paper for publication.
Detailed report.
The authors have tried to write a self consistent paper, but I have the impression that it could be improved by adding some pedagogical sentences here and there. I provide here some personal suggestions in that direction.
I find that the strategy of presenting the results after an historical review make the reading more difficult, I would suggest starting with an introduction to the model and then a section with the main result and the full phase diagram of Fig. 7. This would allow explaining the nature of the various phases before. I would postpone the review about what was already known prior to this paper and how the various phases have been identified.
Specific remarks following the structure of the paper
-Introduction
There is quite a bit of jargon, I would expect the introduction to be accessible to non-experts, though this is not a requirement, but I would suggest to avoid jargon as much as possible at this stage and try to properly define everything.
E.g. what do the authors mean for classical ground states at the beginning of pag. 2? Do they refer to configurations of three dimensional vectors, one per site (e.g. product states).
After or before Eq. 1 I would define the matrices S_i.
I would briefly define what a nematic phase means in this context something like: a phase in which the spin rotational symmetry is broken to rotations of arbitrary angles around one axis, and of pi along any axis in the perpendicular plane (if my uderstanding of the subsequent section is correct).
In particular, I do not see the connection with the nematic phases I am familiar with in liquid crystals or superconductors where I understand them as a breaking of the spatial rotational symmetries rather than the spin rotational symmetries.
-The model
At the end of pag. 4 the quadrupolar states are defined as those for which the quadrupolar moment does not vanish, I would add a definition of the quadrupolar moment.
In the following discussion I would mention that these states will be later represented as circles in the Figures such as Fig. 3, if I have understood correctly.
I would define what a time reversal operator does, this would allow to understand why the base introduced in Eq. 2 is time-reversal invariant.
I would mention after Eq. 3 that the Gell-Mann matrices are the generators of SU(3) or at least provide the relevant references.
In Eq. 4 I would directly insert J_S and J_Q and define them below as already done.
I would explain why the form of the Hamiltonian in terms of the Gell-Mann matrices is SU(3) invariant. A sentence like, an element of SU(3) is represented by a vector in a 8 dimensional space spanned by the Gell-Mann matrices, as a consequence the norm of the vector is invariant under rotations inside this space, representing SU(3) operators... (or something along this line).
-Previous studies
I would move the section further down the paper since it is a review of old results.
In Fig. 2 I would mark the SU(3) invariant points.
End of pag. 6, I would avoid referring to a part of the circle as a "corner", a circle does not have corners, maybe one could say the upper right quadrant?
Figure 3, I would expand the label explaining what the symbols and colours mean. (Maybe one could add a plot with the types of symbols (if I understand correctly the circles are quadrupolar configurations the ovals are nematic, and arrows are magnetic... but what is the colour encoding, x,z,y?).
At the end of pag. 7 3\pi /2 is listed as an SU(3) symmetric point while at pag. 5 it is not. I would add it at pag. 5.
The choice of the determinant of Q as an invariant seems strange since it would vanish as soon as one of the eigenvalues vanishes, that I do not feel means that there is no quadrupolar order, am I wrong? IIQ seems more reasonable. But I would have chosen the average of the abs of the eigenvalues, and their dispersion, in order to provide a more direct information about the magnitude of Q.
What is the advantage of the choice made in the paper?
At pag. 9 the authors mention a hint from the simple update of the existence of the AFM3, can they elaborate more (as done in the following page) or alternatively postpone the discussion completely.
-Description of all the Phases.
Second line I think is opposite directions rather than direction.
Second last line there is an extra to.
In the discussion at pag. 10 there is a part in which it is claimed that the results of the simulations with a unit cell 1x1 give lower energies than larger unit cells, how is it possible?
Is it a sign that the results with larger unit cells are not appropriately converged?
Half through page 10 I would change further investigation to further investigations.
The main results of the paper are presented in the second last paragraph of pag. 12, I would try to bring it forward as already mentioned.
#### Only important question that I expect to be answered
One of the authors (PC) has shown in recent papers that the energy of the iPEPS wave functions can be significantly improved by using new variational minimization algorithm [arXiv:1605.03006 and arXiv:1606.09170v2]. I wonder why such new algorithms have not been applied in the present scenario. It seems important in view of the presence of competing states belonging to different phases whose energy determines the phase of the system. I wonder if the authors have evidences that the variational minimization improves uniformly all competing states, or if they can certify that even a variational optimization would not further improve the energies found in the present study. It seems like a natural test, that could have been performed at least on a couple of points.
#####
A final question concerns the extrapolation performed in Figs. 9-14. From Ref. [49] and Fig. 8 I would have expected an extrapolation as a function of the truncation error. I guess that the data are clear enough that this would not make any difference, but I am curious to understand why the authors have chosen to extrapolate as a function of D, after funding a better strategy. Is it related to the lack of scaling of Q as a function of the truncation error?
I hope that the above observations will help improving the already nice manuscript.
Requested changes
Convincing comments about weakness 1).
Author: Ido Niesen on 2017-10-03 [id 178]
(in reply to Report 1 on 2017-08-24)
We would like to thank the referee for going over our manuscript in such great detail, and for providing many concrete suggestions for improvements. We have tried to decrease the amount of jargon as per the referee's request. A detailed reply to the referee's questions and suggestions can be found below.
First referee: "Detailed report.
The authors have tried to write a self consistent paper, but I have the impression that it could be improved by adding some pedagogical sentences here and there. I provide here some personal suggestions in that direction.
I find that the strategy of presenting the results after an historical review make the reading more difficult, I would suggest starting with an introduction to the model and then a section with the main result and the full phase diagram of Fig. 7. This would allow explaining the nature of the various phases before. I would postpone the review about what was already known prior to this paper and how the various phases have been identified."
We agree with the referee that the results appear quite late in the paper. However, changing the order of the topics does have its disadvantages. For one, it is customary when studying a strongly correlated system to start with the simpler classical (= product state) picture, and then investigate the effects of quantum fluctuations on top of the classical solution. Moreover, we view the historical overview as part of the introduction to the model, because it allows for the different phases to be put into context---rather than displaying a seven-phase phase diagram directly out of the blue---and to explain which part of the phase diagram is interesting, which serves as a motivation for why we focus mostly on the top right octant.
Specifically, we think that discussing the SU(3)-symmetry points should be done before discussing our results, as it explains the location of four out of the seven phase transitions we observe. The discussion on the SU(3)-symmetry points leads very naturally to the discussion of the product state solution, because it can be understood completely in terms of the SU(3)-symmetry points. Moreover, we also refer to the product state solutions multiple times throughout the results section (when discussing the m=1/2 phase, for example). Therefore, we believe the product state discussion should also be placed before the results section. Regarding the remaining three paragraphs of the 'Previous studies' section: moving them we think will not make the paper more readable, because they will be very much out of place at the end of the paper, and they are only three paragraphs, so it won't make that much of a difference.
Therefore, we have decided to stick with the current structure. However, we have added an explicit reference to the full phase diagram to the introduction to guide the more result-oriented reader through the paper. We have also implemented (almost) all other structural/pedagogical recommendations made by the referee.
Referee: "Specific remarks following the structure of the paper
-Introduction
There is quite a bit of jargon, I would expect the introduction to be accessible to non-experts, though this is not a requirement, but I would suggest to avoid jargon as much as possible at this stage and try to properly define everything."
Agreed, and many thanks to the referee for pointing this out. We have tried our best to elaborate on all the points pointed out by the referee.
Referee: "E.g. what do the authors mean for classical ground states at the beginning of pag. 2? Do they refer to configurations of three dimensional vectors, one per site (e.g. product states)."
Indeed, we refer to a lowest energy site-factorized product state as a classical ground state (one that is not entangled). A footnote has been added for clarification.
Referee: "After or before Eq. 1 I would define the matrices S_i."
As is mentioned in the text, the Hamiltonian consists of the "ordinary Heisenberg coupling", and the same coupling squared. We assumed that this would be self-explanatory, but we realize it might not be for everyone. The matrices are the well-known spin matrices, but then for spin-1 particles (hbar = 1). We've added the definitions as requested, and rephrased the whole paragraph to make the description of the model more explicit.
Referee: "I would briefly define what a nematic phase means in this context something like: a phase in which the spin rotational symmetry is broken to rotations of arbitrary angles around one axis, and of pi along any axis in the perpendicular plane (if my understanding of the subsequent section is correct)."
We follow Andreev's [Sov. Phys. JETP 60, 267 (1984)] definition of spin nematic order, which is: spontaneous breaking of the spin group 0(3) while preserving invariance under time reversal. In particular, this means that there can be no on-site magnetic moment (as that would get flipped under time reversal, in the same way that a magnetic dipole originating from a small current loop would flip because the current changes direction under time reversal.) In our case, the preservation of time-reversal symmetry, or rather the absence of magnetic order, comes with additional spatial symmetries.
We have added a sentence to the introduction that explains the notion of a spin-nematic state, as well as a paragraph in Section 3 elaborating on spin-nematic order as defined above and how it relates to quadrupolar states.
Referee: "In particular, I do not see the connection with the nematic phases I am familiar with in liquid crystals or superconductors where I understand them as a breaking of the spatial rotational symmetries rather than the spin rotational symmetries."
As stated above, for spin systems, nematic order occurs when spin-rotational symmetry is broken in the absence of magnetization. An example of a nematic spin-1 single-particle state is the |0> state in the z-basis. Such states can be associated with a director, which is any normal vector to the plane of spin fluctuations (in this case the x-y plane). Unlike magnetic states (described by a magnetic moment, which is not equal to minus itself), nematic states have the property that a director and minus that director both correspond to the same quantum state; that is, nematic single-particle states have the same symmetries as a rod.
Now, the correspondence between liquid crystal nematic and spin-nematic states is that, e.g., in the ferroquadrupolar case, the directors all align along the same direction in the same way rods do in liquid crystal nematic phases, breaking spatial rotation symmetry in the process.
We have added a few sentences to the paper explaining the connection to liquid crystals.
Referee: "-The model
At the end of pag. 4 the quadrupolar states are defined as those for which the quadrupolar moment does not vanish, I would add a definition of the quadrupolar moment."
We have moved the definition of the quadrupolar operators up, and restructured the remaining text accordingly.
Referee: "In the following discussion I would mention that these states will be later represented as circles in the Figures such as Fig. 3, if I have understood correctly."
Done. We've also added a new figure together with an explanation of the meaning of the circles.
Referee: "I would define what a time reversal operator does, this would allow to understand why the base introduced in Eq. 2 is time-reversal invariant."
Included right after the definition of the time-reversal invariant basis in equation 3, followed by an additional discussion on the time-reversal invariance of quadrupolar states.
Referee: "I would mention after Eq. 3 that the Gell-Mann matrices are the generators of SU(3) or at least provide the relevant references."
Done.
Referee: "In Eq. 4 I would directly insert J_S and J_Q and define them below as already done."
Done.
Referee: "I would explain why the form of the Hamiltonian in terms of the Gell-Mann matrices is SU(3) invariant. A sentence like, an element of SU(3) is represented by a vector in a 8 dimensional space spanned by the Gell-Mann matrices, as a consequence the norm of the vector is invariant under rotations inside this space, representing SU(3) operators... (or something along this line)."
We have changed the text to explicitly mention that the Gell-Mann matrices are the generators of SU(3), and added a footnote to elaborate on the point of SU(3) invariance.
Referee: "-Previous studies
I would move the section further down the paper since it is a review of old results."
See above.
Referee: "In Fig. 2 I would mark the SU(3) invariant points."
Done.
Referee: "End of pag. 6, I would avoid referring to a part of the circle as a "corner", a circle does not have corners, maybe one could say the upper right quadrant?"
Good suggestion. We have settled for 'upper right octant', or 'upper right quadrant' depending on which part of the phase diagram we are referring to.
Referee: "Figure 3, I would expand the label explaining what the symbols and colors mean. (Maybe one could add a plot with the types of symbols (if I understand correctly the circles are quadrupolar configurations the ovals are nematic, and arrows are magnetic... but what is the color encoding, x,z,y?)."
We have realized that the meaning of the discs is not self-explanatory. We trust that the addition of the new figure (Fig 2) explains that the discs represent the spin fluctuations, and that a director is a normal vector to this disc of fluctuations. Together with the sentence in the caption of (now) figure 4 which says that the AFQ3 state has neighboring directors being perpendicular (i.e. neighboring discs of fluctuations are such that their normals are perpendicular), we hope that this clarifies the product state picture. In particular, the oval discs are just circular discs viewed from another angle (we have added this comment to the caption of figure 4).
Regarding the color coding: this is only supposed to make it easier for the eyes to see which sites are in the same single-particle state, i.e. to distinguish the iPEPS unit cell (3x3) from the actual unit cell representing the state (3x1), see also footnote [51]. We have added a comment in the caption to clarify this point.
Referee: "At the end of pag. 7 3\pi /2 is listed as an SU(3) symmetric point while at pag. 5 it is not. I would add it at pag. 5."
Thanks for pointing out this oversight. The SU(3)-symmetric point at 3pi/2 was mentioned on p5, but only implicitly (as an element of the set of points for which J_S = - J_Q). We have added the corresponding list of SU(3)-symmetric points to the main text explicitly.
Referee: "The choice of the determinant of Q as an invariant seems strange since it would vanish as soon as one of the eigenvalues vanishes, that I do not feel means that there is no quadrupolar order, am I wrong? IIQ seems more reasonable. But I would have chosen the average of the abs of the eigenvalues, and their dispersion, in order to provide a more direct information about the magnitude of Q.
What is the advantage of the choice made in the paper?"
Indeed, if one of the eigenvalues of Q vanishes this does not mean that there is no quadrupolar order, since quadrupolar order is described by all eigenvalues of Q, not just by one of them. Similarly, one of the matrix invariants being zero does not imply the absence of quadrupolar order. In the same spirit, the 'magnitude of Q' is not a well-defined notion, as Q is not a vector. Rather, it seems more fitting to think of Q as having several types of magnitudes, each of which can be zero or non-zero independently (though they are never all zero simultaneously for pure single-particle states).
For as far as we know, there is no general agreement on which matrix invariants to use (some authors don't even use matrix invariants, but instead just pick one component of Q). We settled for the matrix invariants I-III because they are well-known (they are the coefficients of the characteristic equation of Q) and contain the exact same information as the spectrum of Q, but with the added benefit that one of them (the trace I_Q) is zero by default, which means that all the relevant information is contained in the other two invariants; i.e. plotting the matrix invariants is more economical than plotting all the eigenvalues.
We could have also used the average of the absolute values of the eigenvalues and their spread, but, if we are not mistaken, these two numbers are not in one-to-one correspondence with the sets of three eigenvalues summing up to zero (i.e., due to the absolute value, which forgets about the sign, different sets of eigenvalues can have the same average of their absolute values and spread), and we prefer to work with quantities that uniquely identify the quadrupolar order of the state in question.
Referee: "At pag. 9 the authors mention a hint from the simple update of the existence of the AFM3, can they elaborate more (as done in the following page) or alternatively postpone the discussion completely."
The iPEPS simulations at \theta=0.2\pi show that the state with the 3x3 unit cell is competitive in energy, and all of these 3x3-unit-cell simulations display the AFM3 magnetization pattern displayed in figure 6. The actual ground state sits in the Haldane phase and is not magnetized, but it does not show itself when using low bond dimension simulations with simple updates (which we did in Section 5.1).
We have added a footnote to elaborate on the statement in question.
Referee: "-Description of all the Phases.
Second line I think is opposite directions rather than direction."
Indeed, thanks!
Referee: "Second last line there is an extra to."
And thanks for this as well!
Referee: "In the discussion at pag. 10 there is a part in which it is claimed that the results of the simulations with a unit cell 1x1 give lower energies than larger unit cells, how is it possible?
Is it a sign that the results with larger unit cells are not appropriately converged?"
Not exactly. The larger unit cell simulations are converged, but in a particular state.
What we observe is the following: comparing simulations for fixed but low bond dimension D, the larger unit cell iPEPS are lowest in energy. However, as we increase D, the 1x1 unit cell iPEPS becomes more and more competitive, and from about D~8 (depending on \theta) and onward it actually attains the lowest energy (for fixed D) and remains lowest in energy as we extrapolate D to infinity.
When increasing the bond dimension, we initialize each fixed D simulation from a simulation with a bond dimension of D-1, and in this particular case the low D lowest energy iPEPS happens to be magnetized. That is to say that, although the actual ground state lies in the Haldane phase, when we force the state to be slightly entangled (low D), the lowest energy is obtained by allowing the state to develop on-site magnetization. Note that this only happens for larger unit cell iPEPS, as a 1x1 unit cell is not compatible with AFM-like translational symmetry breaking. However, as we increase D, the larger unit cell iPEPS remain stuck in a magnetic state. This is a consequence of the fact that both simple and full updates locally update one bond in the unit cell at the time, allowing for simulation to stay stuck close to a local (magnetic) energy minimum that is not the global minimum (Haldane). Note that a similar situation can also happen close to a phase transition when a certain degree of hysteresis can be observed. Only when going to higher D do we observe for the larger unit cell iPEPS that the magnetization decreases (and eventually extrapolates) to zero.
In short, the larger unit cell simulations start off in the 'wrong' low D state, and remain close to it as we ramp up D, whereas the 1x1 unit cell iPEPS goes directly towards the actual ground state of the system as D increases. Technically, whenever the ground state preserves translational symmetry, it should of course be possible to obtain the exact same energy with 1x1 or a larger unit cell iPEPS. The observed difference in energies is really a remnant of the lower D iPEPS' preference for developing magnetization combined with the way we initialize our simulations.
Referee: "Half through page 10 I would change further investigation to further investigations."
This is a matter of style. In this case, 'further investigation' is meant to be read as 'continued investigation', in which case the singular form is appropriate. We could have also used the plural, but, if the referee does not mind, we would prefer to leave it as it is.
Referee: "The main results of the paper are presented in the second last paragraph of pag. 12, I would try to bring it forward as already mentioned."
See above.
Referee: "#### Only important question that I expect to be answered
One of the authors (PC) has shown in recent papers that the energy of the iPEPS wave functions can be significantly improved by using new variational minimization algorithm [arXiv:1605.03006 and arXiv:1606.09170v2]. I wonder why such new algorithms have not been applied in the present scenario. It seems important in view of the presence of competing states belonging to different phases whose energy determines the phase of the system. I wonder if the authors have evidences that the variational minimization improves uniformly all competing states, or if they can certify that even a variational optimization would not further improve the energies found in the present study. It seems like a natural test, that could have been performed at least on a couple of points."
The answer to this question is twofold. First, it is very important to understand that the full update method is systematic (unlike the simple update approach), i.e. it yields a systematic improvement with increasing bond dimension and converges to the true ground state in the large D limit. Our conclusions are based on extrapolated full update results (not fixed D results) and are therefore reliable. It is true that the variational optimization can improve the results for fixed bond dimension, but it should eventually converge to the same result (i.e. it converges to the exact result more quickly than the full update).
Second, and this is the main reason we have not used the variational optimization in this work, the variational optimization algorithm is still under development and we currently only have a prototype code. In its current version it requires significantly more memory than the full update so that the latter can currently more easily be pushed to large bond dimensions, especially also when using the recently developed fast-full update. And having a large range of data points is important for an accurate extrapolation, i.e. we can do a more reliable extrapolation using the full-update data points up to large D than with only a few low-D data points from the variational optimization. We are currently working on introducing further tricks in the variational approach in order to reduce memory costs, but this still requires further testing and benchmarking, and is beyond the scope of this paper.
In short: given the current state of the variational optimization algorithm, for the spin-1 BBH model we can obtain more reliable extrapolated D to infinity results using the full update, which is why we stuck with the latter for this paper.
Referee: "A final question concerns the extrapolation performed in Figs. 9-14. From Ref. [49] and Fig. 8 I would have expected an extrapolation as a function of the truncation error. I guess that the data are clear enough that this would not make any difference, but I am curious to understand why the authors have chosen to extrapolate as a function of D, after funding a better strategy. Is it related to the lack of scaling of Q as a function of the truncation error?"
The truncation error extrapolation has only been been properly benchmarked on the energy per site, and we have encountered situations (with other models) where truncation error extrapolation does not work well for order parameters. So, to be on the safe side, we stuck with 1/D extrapolation. It should be noted that truncation error extrapolation of the magnetization and quadrupole eigenvalues do provide qualitatively the same answer: i.e. we observe the same jumps in the order parameters for all phase transitions investigated in the paper.
Referee: "I hope that the above observations will help improving the already nice manuscript."
Thanks again for the extensive list of suggested improvements!
Author: Ido Niesen on 2017-10-03 [id 179]
(in reply to Report 2 on 2017-08-24)We want to thank the referee for his or her comment and interesting question.
The Haldane phase we find in the two-dimensional spin-1 BBH model on the square lattice is not adiabatically connected to the two-dimensional AKLT state that can be obtained by taking four spin-1/2 particles per lattice site and projecting onto their combined spin-2 subspace, thereby forming a lattice of spin-2 spins. This two-dimensional AKLT state also does not break lattice rotation symmetry, which is something that we clearly do observe in our simulations. Rather, what we claim is that the phase we find can be adiabatically connected to the state of decoupled one-dimensional ALKT chains (i.e. infinitely many copies of them, one next to the other, so that together they form a two-dimensional grid of one-dimensional chains) by gradually turning off the coupling in either x or y direction. In terms of valance bonds: they will form only in one direction, so only two spin-1/2 particles per site are required before projection (onto the on-site spin-1 subspace), not four.
As we described in our previous paper [DOI: 10.1103/PhysRevB.95.180404], our investigation of the phase diagram of the anisotropic BBH model displays a clear uninterrupted path from the one-dimensional Haldane phase to the two-dimensional Haldane phase, showing that the two are adiabatically connected. This notion is reinforced by the fact that states in the two-dimensional Haldane phase break lattice rotation symmetry in such a way that the bond energies in one direction are lower than in the other, while preserving spin-rotation and lattice translation symmetries, mimicking the structure of decoupled one-dimensional AKTL chains.
We hope this answers the possible confusion about the relationship between the two-dimensional Haldane phase and the two-dimensional AKLT state. In short: they are not the same. We have added an extra paragraph to the paper to clarify this point.

---

## Round 3 · Author Response

Dear editor,

Thank you for forwarding the referee reports on the manuscript titled "A tensor network study of the complete ground state phase diagram of the spin-1 bilinear-biquadratic Heisenberg model on the square lattice".

Both referees are positive about the paper and recommend publication after we address the points discussed below.

The first referee's concern about the absence of the use of variational optimization will be discussed in our direct reply to the first referee. Additionally, the same reply contains an extensive pointwise discussion on the referee's excellent suggestions for improvements regarding the amount of jargon in the paper. The list of changes made in accordance to the referee's suggestions can be found below. The second referee's concern about how the two-dimensional AKLT state relates to the Haldane phase we find in the BBH model will be addressed in our direct reply to the second referee. In addition, we have added an extra paragraph to the paper clarifying the issue raised by the referee.

We hereby submit a revised version of our paper containing the changes listed below to SciPost.

Sincerely yours,
Ido Niesen and Philippe Corboz

---

## Round 3 · List of Changes

List of changes:

  1. Introduction:
  2. added a footnote explaining the term "classical ground state",
  3. extended a sentence to explain the meaning of spin-nematic states,
  4. rewrote the second paragraph to include a more explicit description of Hamiltonian and corresponding operators, and,
  5. added an explicit reference to the iPEPS phase diagram.

  6. The model: rewrote most of this section to make it more accessible to non-experts. In particular, we

  7. moved the definition of the quadrupolar operators forward,
  8. added a figure explaining the meaning of the discs (quadrupolar states),
  9. defined the action of the time-reversal operator explicitly, and elaborated on the notion of spin-nematic states (in relation to time-reversal invariance),
  10. added a few sentences on the relation between spin-nematics and nematics in liquid crystals,
  11. inserted J(S) and J(Q) directly into Eq. 4,
  12. explicitly listed all SU(3)-symmetric points, and,
  13. added a sentence and a footnote on the Gell-Mann matrices and their relation to SU(3).

  14. Previous studies:

  15. marked the SU(3) points with black dots in Fig. 3,
  16. added note on the color coding in Fig. 4, and,
  17. changed 'corner' to 'octant'.

  18. Description of all phases:

  19. added a paragraph explaining that the Haldane phase we find does not correspond to the two-dimensional AKLT state.

---

## Editorial Decision

published